Diagnostic performance of circulating tumor DNA as a minimally invasive biomarker for hepatocellular carcinoma: a systematic review and meta-analysis

Li Jia Jie 1
Lv Yanqing 2
Ji Huifan jihf@jlu.edu.cn 3
1 Hepatobiliary Pancreatic Department, The First Hospital of Jilin University , Changchun , Jilin , China
2 Department of Hepatobiliary and Pancreatic Medicine, The First Hospital of Jilin University , Changchun , Jilin , China
3 Department of Hepatobiliary and Pancreatic Medicine, The First Hospital of Jilin University , Changchun , Jilin , China
Chen Gang
Electronic publication date: 2022 Nov 3
Publication date: 2022
Volume: 10
Electronic Location ID: e14303
Received 2022 May 5; Accepted 2022 Oct 5
Copyright: ©2022 Li et al.
Copyright year: 2022
Copyright holder: Li et al.
License: This is an open access article distributed under the terms of the Creative Commons Attribution License, which permits unrestricted use, distribution, reproduction and adaptation in any medium and for any purpose provided that it is properly attributed. For attribution, the original author(s), title, publication source (PeerJ) and either DOI or URL of the article must be cited.
License URL: https://creativecommons.org/licenses/by/4.0/

Keywords: Circulating tumor DNA, Hepatocellular carcinoma, DNA Methylation, mSEPT9, Meta-analysis, Diagnostic performance

Funding: Jilin Provincial Health Special Project The authors received funding from the Jilin Provincial Health Special Project. The funders had no role in study design, data collection and analysis, decision to publish, or preparation of the manuscript.

==============================
Purpose

This study aimed to assess the diagnostic performance of circulating tumor DNA (ctDNA) in hepatocellular carcinoma (HCC).

Materials and Methods

We enrolled all relevant studies published up to 5 January 2022. Three primary subgroups were investigated: qualitative or quantitative ctDNA analyses, combined alpha-fetoprotein (AFP), and ctDNA assay. In addition to the three primary subgroups, we also evaluated the diagnostic value of methylated SEPTIN9 (mSEPT9), which has been studied extensively in the diagnosis of hepatocellular carcinoma. After a search based on four primary databases, we used a bivariate linear mixed model to analyze the pooled sensitivity (SEN), specificity (SPE), positive likelihood ratio (PLR), negative likelihood ratio (NLR), and diagnostic odds ratio (DOR). We also plotted hierarchical summary receiver operating characteristics (HSROC) and utilized lambda as well as the area under the curve (AUC) to create summary receiver operating characteristic (SROC) curves to estimate the diagnostic value of ctDNA.

Results

A total of 59 qualified articles with 9,766 subjects were incorporated into our meta-analysis. The integrated SEN, SPE, and DOR in the qualitative studies were 0.50 (95% CI [0.43–0.56]), 0.90 (95% CI [0.86–0.93]), and 8.72 (95% CI [6.18–12.32]), respectively, yielding an AUC of 0.78 and lambda of 1.93 (95% CI [1.56–2.33]). For quantitative studies, the corresponding values were 0.69 (95% CI [0.63–0.74]), 0.84 (95% CI [0.77–0.89]), 11.88 (95% CI [7.78–18.12]), 0.81, and 2.32 (95% CI [1.96–2.69]), respectively. Six studies were included to evaluate the SETP9 methylation, which yielded an AUC of 0.86, a SEN of 0.80 (95% CI [0.71–0.87]), and a SPE of 0.77 (95% CI [0.68–0.85]). Likewise, ctDNA concentration yielded an AUC of 0.73, with a SEN of 0.63 (95% CI [0.56–0.70]) and a SPE of 0.86 (95% CI [0.74–0.93]). AFP combined with ctDNA assay resulted in an AUC of 0.89, with a SEN of 0.82 (95% CI [0.77–0.86]) and a SPE of 0.84 (95% CI [0.76–0.90]).

Conclusion

This study shows that circulating tumor DNA, particularly mSEPT9, shows promising diagnostic potential in HCC; however, it is not enough to diagnose HCC independently, and ctDNA combined with conventional assays such as AFP can effectively improve diagnostic performance.

Introduction

According to the World Health Organization’s 2020 report, liver cancer is ranked as the sixth most common tumor type and the third most common cause of cancer-related deaths worldwide, with more than 905,000 new cases and 577,522 deaths in 2020 (https://gco.iarc.fr/today/fact-sheets-cancers). Among all primary liver cancers, hepatocellular carcinoma (HCC) accounts for >80% of primary liver cancers worldwide (El-Serag & Rudolph, 2007). Although many HCC treatments are available, such as local ablation, surgical resection, and liver transplantation, the majority of patients have poor prognoses due to the fact that they are diagnosed and treated at the late stages of HCC. Currently, abdominal ultrasonography and serum alpha-fetoprotein (AFP) measurement are widely accepted as the most effective and affordable tools in clinical work. The diagnostic performance of existing tumor biomarker tests is relatively low when screening for HCC, with a sensitivity (SEN) of 0.478 (95% CI [0.447–0.509]) and a specificity (SPE) of 0.840 (95% CI [0.809–0.867]) (Zhang et al., 2020b) for AFP assay, and imaging technology typically only detect tumors that are greater than one cm in diameter (Maluccio & Covey, 2012). Therefore, it is imperative to develop novel non-invasive biomarkers that are more sensitive at the early stages of liver cancers and can overcome the shortcomings of conventional biomarkers.

Over the past 10 years, liquid biopsy has attracted substantial attention as a supplement or alternative biomarker to conventional biomarkers (AFP, AFP-L3, and PIVKA-II) and tissue biopsy for tumor diagnosis and monitoring (Chen & Zhao, 2019; Crowley et al., 2013; Kondo, Kimura & Shimosegawa, 2015). Liquid biopsy is a minimally invasive procedure that usually samples blood, cerebrospinal fluid, urine, sputum, ascites, or theoretically any other body fluid (Dell’Olio et al., 2020). Liquid biopsy initially analyzed only circulating tumor cells (CTCs), but now extends to the analysis of the many components released by the tumor in body effluents (mainly blood), including cell-free circulating DNA, mRNA, non-coding RNA, long non-coding RNA, glycoprotein, “tumor educated platelets” (TEPs), or vesicles such as exosomes (Poulet, Massias & Taly, 2019). Our focus is on the unique entity of ctDNA in blood, which exhibits the heterogeneity of primary tumors and offers the potential of being used to detect or monitor tumors in patients without obvious clinical diseases (Corcoran & Chabner, 2018).

In 1977, Leon et al. (1977) reported that many cancer patients had elevated circulating cell-free DNA (ccfDNA). The quantity of the ccfDNA associated with disease burden indicated that some of these DNA were of tumor origin (Leon et al., 1977). From the blood of cancer patients, a portion of the cfDNA. was released by tumor cells through apoptosis, necrosis, or active release (Stroun et al., 2001), and these cells carried cancer-specific gene or epigenetic modifications, including single nucleotide mutation (Huang et al., 2003), copy number aberration (can) (Allen Chan et al., 2013), DNA methylation (Wang et al., 2021), 5-hydroxymethylcytosines (Zhang et al., 2020b; Cai et al., 2019), and cfDNA integrity (Huang et al., 2016). As more advanced molecular biology techniques, such as next-generation sequencing, developed, scientists were able to monitor and diagnose cancers through quantitative and qualitative analysis of ctDNA. DNA methylation, the most important epigenetic modification, is considered a promising tool for cancer diagnosis (Zhang et al., 2019) and has been considered as a novel discriminatory tool for the screening, detection, and diagnosis of HCC over the past decade. Although the accuracy of circulating ctDNA assays in the detection of HCC has been previously reported, the results were distinctly different. A systematic review and meta-analysis assessing the diagnostic performance of ctDNA assays in HCC was published two years ago (Zhang et al., 2020b), and we have decided to explore this subject further due to the following reasons: a number of additional studies on the correlation between ctDNA and HCC diagnosis have been published that will allow a more comprehensive synthesis of the corresponding data; there are more studies that the previous meta-analysis did not mention that we enrolled in this study; and we also analyzed the diagnostic value of methylated SEPTIN9 (mSEPT9), which was approved as the first blood-based early detection test for colorectal cancer and was reported recently to be a promising biomarker for diagnosing HCC in Chinese and European patients with cirrhosis.

Materials and Methods

Search strategy

Two authors (Jiajie Li and Yanqing Lv) independently conducted a comprehensive search for relevant articles in PubMed, Web of Science, Embase, and Cochrane Library. The query terms were as follows: “circulating tumor DNA” OR “circulating DNA” OR “ctDNA” OR “plasma DNA” OR “serum” DNA” OR “blood DNA” and “Liver Neoplasms” OR “Neoplasms, Hepatic” OR “Neoplasms, Liver” OR “Liver Neoplasm” OR “Neoplasm, Liver” OR “Hepatic Neoplasms” OR “Hepatic Neoplasm” OR “Neoplasm, Hepatic” OR “Cancer of Liver” OR “Hepatocellular Cancer” OR “Cancers, Hepatocellular” OR “Hepatocellular Cancers” OR “Hepatic Cancer” OR “Cancer, Hepatic” OR “Cancers, Hepatic” OR “Hepatic Cancers” OR “Liver Cancer” OR “Cancer, Liver” OR “Cancers, Liver” OR “Liver Cancers” OR “Cancer of the Liver” OR “Cancer, Hepatocellular” AND “diagnosis” OR “sensitivity” OR “specificity” OR “accuracy”. The language of all the articles was limited to English and the experiment target was limited to human beings. We also manually screened references from included articles and related reviews in order to expand the search.

Inclusion and exclusion criteria

Studies that met the following criteria were eligible for inclusion: (a) diagnostic accuracy of ctDNA was evaluated in plasma or serum; (b) sufficient data were acquired or could be calculated from the raw data (e.g., SEN, SPE, true positives [TP], false positives [FP], true negatives [FN], and false negatives [FN]); (c) CtDNA markers were used for the first diagnosis of HCC, not for the diagnosis of recurrence and metastasis; (d) controls were cancer-free adults; and (e) articles were published in English and experiment sources were human beings. The main exclusion criteria were as follows: (a) reviews, conference abstracts, meta-analysis, editorials, letters, reply, case report, commentary, short survey, notes, research highlight, and duplicate publications; (b) the sample size of studies was less than 10; (c) we failed to obtain the full text; (d) overlapping publications that included the same population and gene; and (e) there were multiple genes or gene models.

Data extraction & quality assessment

The following data from the eligible data were extracted by two reviewers independently: the first authors’ name, publication year, country or region of origin, study design, sampling time, inspection method, assay indicator, cut-off values, number of participants, and SP, SE, TP, FP, FN, and TN, which were given directly or could be calculated by raw data. Also, based on the revised Quality Assessment of Diagnostic Accuracy Studies-2 (QUADAS-2), the quality of studies was assessed and rated by two authors independently, and the risk of bias and applicability concerns was categorized as low, unclear, or high. If the answer to all the iconic questions in a range was “yes”, then the risk of bias can be assessed as low, if the answer to any of the information questions is “no”, then the risk of bias was judged as “high”, and when there was not enough information, we defined it as unclear risk. Divergences were discussed together to reach a consensus and if an agreement was unable to be met, Huifan Ji made a judgement.

Statistical analysis

We used Stata software (version 16.0; Stata Corporation, TX, USA), meta-disc 1.4, to perform this diagnostic meta-analysis. The pooled sensitivity and specificity, positive likelihood ratio (PLR), negative likelihood ratio (NLR), diagnostic odds ratio (DOR), and corresponding 95% confidence interval (95% CI) were calculated using a bivariate generalized linear mixed model to evaluate the test accuracy. Simultaneously, we plotted hierarchical summary receiver operating characteristics (HSROC) and utilized lambda as well as the area under the curve (AUC) which created by summary receiver operating characteristic (SROC) curves to estimate the diagnostic value of ctDNA. The logit estimates of SEN, SPE, and respective variances were used to construct the HSROC curves. Fagan’s nomogram was applied to interpret the clinical utility of ctDNA for HCC (Anthony, 2007). The Spearman correlation coefficient and its corresponding P value were used to identify the presence of the threshold effect. The threshold effect was considered to exist when the P value was lower than 0.05. If heterogeneity resulted from the non-threshold effect, I2 and χ2 were used to evaluate heterogeneity between the enrolled articles. I2 >50% or P < 0.05 for χ2 suggested significant heterogeneity (Borenstein et al., 2017). Subgroup and meta regression analyses were used to explore the sources of heterogeneity. Finally, we formulated Deek’s funnel plot to examine potential publication bias (Deeks, Macaskill & Irwig, 2005), and trim and fill analysis was applied to assess the effect of bias on the pooled estimate once the publication bias existed.

Results

Study characteristics

Figure 1 shows the process of literature retrieval and inclusion. Through our search strategy, a total of 879 publications were enrolled from different databases outlined in the Materials and methods section. After deleting 216 duplicated articles, 487 articles were excluded after screening their titles and abstracts (Fig. 1). Eventually, the full text of 59 eligible articles (60 studies) was incorporated into this meta-analysis (Huang et al., 2003; Wang et al., 2021; Wong et al., 2000; Wong et al., 2003; Chu et al., 2004; Lin et al., 2005; Yeo et al., 2005; Iizuka et al., 2006; Ren et al., 2006; Wang et al., 2006; Zhang et al., 2007; Chan et al., 2008; Chang et al., 2008; Igetei et al., 2008; Ahmed et al., 2010; El-Shazly et al., 2010; Hu et al., 2010; Huang et al., 2011; Iizuka et al., 2011; Yang et al., 2011; Chen et al., 2012; Huang et al., 2012; Mohamed et al., 2012; Chen et al., 2013; Sun et al., 2013; Zhang et al., 2013; Han et al., 2014; Huang et al., 2014; Ji et al., 2014; Kuo et al., 2014; Li et al., 2014; Yang et al., 2014; Dong et al., 2015; Huang et al., 2015; Ramadan et al., 2015; Teng et al., 2016; Dong et al., 2017; Hu et al., 2017; Mansour et al., 2017; Tian et al., 2017; Wu et al., 2017; Gai et al., 2018; Li et al., 2018; Oussalah et al., 2018; Wei et al., 2018; Linlin et al., 2018; Bai et al., 2019; Kisiel et al., 2019; Marchio et al., 2019; Pasha, Mohamed & Radwan, 2019; El-Bendary et al., 2020; He et al., 2020; Kotoh et al., 2020; Li et al., 2020a; Liu et al., 2020; Qian et al., 2020; Akuta et al., 2021; Lewin et al., 2021; Xie et al., 2021).

Figure 1 A PRISMA flow diagram of the literature search.

CNV, copy number variations; ctDNA, circulating tumor DNA; HCC, hepatocellular carcinoma; PRISMA, Preferred Reporting Items for Systematic Reviews and Meta-Analyses; PBMC, peripheral mononuclear blood cell; SEN, sensitivity; SPE, specificity; SNP, single nucleotide polymorphism.

Baseline characteristics

Table S1 summarizes the characteristics of all 59 papers (n = 92). All the included studies included quantitative analysis to measure ctDNA concentration and single-gene methylation concentration (n = 19) (Wang et al., 2021; Iizuka et al., 2006; Ren et al., 2006; Chan et al., 2008; El-Shazly et al., 2010; Iizuka et al., 2011; Yang et al., 2011; Chen et al., 2012; Huang et al., 2012; Mohamed et al., 2012; Chen et al., 2013; He et al., 2020; Kotoh et al., 2020; Li et al., 2020a; Xie et al., 2021; Mansour et al., 2017; Gai et al., 2018; Linlin et al., 2018; El-Bendary et al., 2020) and qualitative analysis of tumor-specific ctDNA single gene-mutation and methylation (n = 40) (Wong et al., 2000; Wong et al., 2003; Chu et al., 2004; Lin et al., 2005; Yeo et al., 2005; Wang et al., 2006; Zhang et al., 2007; Chang et al., 2008; Igetei et al., 2008; Ahmed et al., 2010; Hu et al., 2010; Huang et al., 2011; Sun et al., 2013; Zhang et al., 2013; Han et al., 2014; Huang et al., 2014; Ji et al., 2014; Kuo et al., 2014; Li et al., 2014; Yang et al., 2014; Dong et al., 2015; Huang et al., 2015; Ramadan et al., 2015; Teng et al., 2016; Dong et al., 2017; Hu et al., 2017; Tian et al., 2017; Wu et al., 2017; Li et al., 2018; Oussalah et al., 2018; Wei et al., 2018; Bai et al., 2019; Kisiel et al., 2019; Marchio et al., 2019; Pasha, Mohamed & Radwan, 2019; Liu et al., 2020; Qian et al., 2020; Akuta et al., 2021; Lewin et al., 2021). Of these papers, 18 articles evaluated the diagnostic performance of ctDNA combined with AFP assay in HCC (Wang et al., 2021; Chan et al., 2008; Huang et al., 2012; Chen et al., 2013; Han et al., 2014; Huang et al., 2014; Kuo et al., 2014; Li et al., 2014; Yang et al., 2014; Dong et al., 2015; Teng et al., 2016; Hu et al., 2017; Tian et al., 2017; Pasha, Mohamed & Radwan, 2019; Kotoh et al., 2020; Li et al., 2020a; Liu et al., 2020; Qian et al., 2020). In the qualitative analysis subgroup, our study enrolled a total population of 3,072 HCC patients and 3,413 control individuals (2,064 patients with benign liver disorders or liver cirrhosis, 1,001 healthy volunteers, and 308 non-cancer controls in a combined group of patients with benign liver disorders and healthy controls). Patients with chronic hepatitis, benign hepatic lesions, and cirrhosis were selected as the control group in 13 articles (Chu et al., 2004; Wang et al., 2006; Chang et al., 2008; Ahmed et al., 2010; Huang et al., 2011; Huang et al., 2014; Ji et al., 2014; Huang et al., 2015; Oussalah et al., 2018; Kisiel et al., 2019; Qian et al., 2020; Akuta et al., 2021; Lewin et al., 2021) , while eight articles with only healthy control groups were chosen in this subgroup (Yeo et al., 2005; Zhang et al., 2007; Igetei et al., 2008; Hu et al., 2010; Zhang et al., 2013; Kuo et al., 2014; Bai et al., 2019; Marchio et al., 2019) . Other articles combined healthy volunteers and chronic hepatitis or liver cirrhosis patients as the control group. The majority of articles were conducted in Asia (n = 31) (Huang et al., 2003; Wong et al., 2000; Wong et al., 2003; Chu et al., 2004; Lin et al., 2005; Yeo et al., 2005; Wang et al., 2006; Zhang et al., 2007; Chang et al., 2008; Hu et al., 2010; Huang et al., 2011; Sun et al., 2013; Zhang et al., 2013; Han et al., 2014; Ji et al., 2014; Kuo et al., 2014; Li et al., 2014; Yang et al., 2014; Dong et al., 2015; Huang et al., 2015; Teng et al., 2016; Dong et al., 2017; Hu et al., 2017; Tian et al., 2017; Wu et al., 2017; Li et al., 2018; Wei et al., 2018; Bai et al., 2019; Liu et al., 2020; Qian et al., 2020; Akuta et al., 2021), five in Africa (Igetei et al., 2008; Ahmed et al., 2010; Ramadan et al., 2015; Marchio et al., 2019; Pasha, Mohamed & Radwan, 2019), three in America (Huang et al., 2014; Kisiel et al., 2019; Lewin et al., 2021) , and one in Europe (Oussalah et al., 2018). A total of 37 studies looked at ctDNA methylation, while three evaluated single-gene mutation. In terms of study type, seven studies were retrospective, four were prospective, and the rest (n = 31) did not clearly state the study design. Among those factors known at the time of collection, sampling time was either before treatment or surgery (n = 19), and samples were obtained from plasma (n = 15), serum (n = 24), or plasma/serum (n = 1). There were 23 articles with sample sizes ≥100, while the remaining sample sizes<100.

In the quantitative analysis subgroup, there were a total of 1,446 HCC patients and 1,835 non-cancer control participants (966 patients, 650 healthy volunteers, and 219 participants in a mixed benign liver disorders and healthy control group). Patients with liver cirrhosis, HCV infection, and HBV infection were chosen as the control group in nine publications (Iizuka et al., 2006; El-Shazly et al., 2010; Iizuka et al., 2011; Chen et al., 2012; Huang et al., 2012; Mohamed et al., 2012; Mansour et al., 2017; Linlin et al., 2018; Kotoh et al., 2020) , and only one study singly chose healthy volunteers as control group (Chen et al., 2013). Among these studies, all but four articles were conducted in Egypt (El-Shazly et al., 2010; Mohamed et al., 2012; Mansour et al., 2017; El-Bendary et al., 2020) , and the others were all studied in Asia (n = 15). Eight studies evaluated the performance of ctDNA concentration as a diagnostic indicator, 10 studies chose single gene methylation concentration, and one selected hTERT concentration. As for study design, the majority of them were not described clearly (n = 16), but two were prospective studies, and one was a retrospective study. Nine studies had a known time of collection before treatment and surgery. Samples were obtained from plasma (n = 9) and serum (n = 10). Twelve publications had sample sizes ≥100 and the rest had sample sizes smaller than 100. The assay methods used to measure the concentrations of ctDNA were real-time quantitative polymerase chain reaction (RT-qPCR) (n = 5), droplet digital PCR DNA (Dd-PCR) (n = 3), quantitative methylation specific PCR (Q-MSP) (n = 4), quantitative PCR (QPCR) (n = 4), ultraviolet transilluminator (n = 1), and Qubit dsDNA (n = 1). For ctDNA mixed with the AFP subgroup, 18 papers were studied with 1,790 HCC patients and 1,614 non-cancer participants.

Quality assessment

As shown in Fig. 2, the majority of enrolled studies included four criteria: patient selection, index test assessment, reference standard assessment, and a flow and timing assessment. Publications were judged as having high risk in one field, which was consider as having an overall high risk of bias. However, two studies were excluded due to the high risk of bias in patient selection and reference concerns (Lleonart et al., 2005; Marchio et al., 2018) , and four articles (Chan et al., 2008; Ahmed et al., 2010; Huang et al., 2012; Lewin et al., 2021) had a high risk of bias regarding patient selection and others had a risk of bias in index text (He et al., 2020) or reference standards (Huang et al., 2003). Additionally, due to missing information, the risk of bias could not be assessed for another 16 studies (Yeo et al., 2005; Iizuka et al., 2006; Igetei et al., 2008; Chen et al., 2013; Li et al., 2014; Yang et al., 2014; Ramadan et al., 2015; Dong et al., 2017; Mansour et al., 2017; Li et al., 2018; Wei et al., 2018; Linlin et al., 2018; Bai et al., 2019; Marchio et al., 2019; El-Bendary et al., 2020; Xie et al., 2021). Because many case-control studies were enrolled in this meta-analysis, the selection of patients was the main bias risk across the included publications.

Figure 2 Quality assessment of the included studies, for each study, risk of bias and applicability concerns were categorized as low, unclear or high.

(A) and (B) Quality assessment of the included studies based on the quality assessment of diagnostic accuracy studies criterion. For each study, risk of bias and applicability concerns categorized as low, unclear or high. (C) Each bar represents the percent of studies considered as high risk, low risk or unclear for both risk of bias and applicability concerns.

Diagnostic performance

Diagnostic accuracy of the quantitative and qualitative analysis of ctDNA for HCC

The qualitative detection of ctDNA discriminated HCC patients from control individuals with a SEN of 0.50 (95% CI [0.43–0.56], I2 statistic: 94.01%) and a SPE of 0.90 (95% CI [0.86–0.93], I2 statistic: 95.84%) (Fig. 3). In addition, the pooled PLR was 4.89 (95% CI [3.66–6.53], I2 statistic:92.21%), NLR was 0.56 (95% CI [0.50–0.63], I2 statistic:95.63%), DOR was 8.72 (95%CI [6.18–12.32], I2 statistic:100%), and the AUC for the SROC curve was 0.78 (95%CI [0.74–0.81]). The HSROC graph was plotted and the value of beta was 0.23 (95% CI [−0.05–0.56]), z value was 1.61, and p value was 0.106, which indicated that the graph was symmetrical. Lambda was 1.93 (95% CI [1.56–2.33]), which suggested a moderate level of diagnostic value (Fig. 4A). The Spearman correlation coefficient was 0.367 and p-value = 0.003, indicating that heterogeneity among studies was derived from non-threshold effects. In the same way, the pooled SE and SP for the diagnostic performance of the quantitative detection in HCC were 0.69 (95% CI [0.63–0.74], I2 statistic:86.32%) and 0.84 (95% CI [0.77–0.89], I2 statistic:92.87%), respectively (Fig. 5). The PLR was 4.36 (95% CI [3.02–6.30], I2 statistic: 88.24%) and NLR was 0.37 (95% CI [0.31–0.43], I2 statistic: 82.48%). The pooled DOR was 11.87 (95%CI [7.78–18.12]; I2 statistic:100%) and the AUC for the SROC curve was 0.81 (95% CI [0.77–0.84]). In addition, the value of beta was 0.62 (95% CI [0.17–1.07]), z statistic was 2.68, and p value was 0.007, indicating that the HSROC was asymmetric. Lambda was 2.32 (95%CI [1.96–2.69]), which indicated a moderate level of diagnostic value (Fig. 4B). The Spearman correlation coefficient of 0.494 and the p value of 0.014 indicated that there was no threshold effect.

Figure 3 Forest plots of SEN and SPE for diagnostic performance of ctDNA assay for HCC in the qualitative detection subgroup.

Figure 4 The hierarchical summary receiver operating characteristic curves.

(A) The diagnostic accuracy of the qualitative subgroup (B)The diagnostic accuracy of the quantitative subgroup. (C) The diagnostic accuracy of the ctDNA combined with AFP subgroup. (D) The diagnostic accuracy of the ctDNA concentration subgroup. (E) The diagnostic value of the SETP9 methylation subgroup. HSROC, Hierarchical summary receiver operating characteristic.

Figure 5 Forest plots of SEN and SPE for diagnostic performance of ctDNA assay for HCC in the quantitative detection subgroup.

Diagnostic performance of ctDNA combined with AFP assay for HCC

Using the combination of ctDNA and AFP as detective indicators distinguished HCC patients from non-cancer control participants with a SEN of 0.82 (95% CI [0.77–0.86], I2 statistic: 85.81%) and a SPE of 0.84 (95% CI [0.76–0.90], I2 statistic:93.32%) (Fig. 6). The combined PLR, NLR, and DOR was 5.13 (95% CI [3.31–7.96], I2 statistic:92.31), 0.22 (95% CI [0.17–0.28], I2 statistic:85.91), 23.63 (95% CI [12.82–23.56]; I2 statistic:100%), respectively. The results obtained by the HSROC model showed that the value of beta was 0.56 (95% CI [−0.03–1.15]), z statistic was 1.87, and p value was 0.062. The estimate for the “Lambda” and its 95% confidence interval was 3.24 (95% CI [2.63–3.85]), suggesting a high level of diagnostic value (Fig. 4C).

Figure 6 Forest plots of SEN and SPE for diagnostic value of ctDNA assay for HCC in the combined ctDNA-AFP assay detection subgroup.

AFP, alpha-fetoprotein; ctDNA, circulating tumor DNA; HCC, hepatocellular carcinoma; SEN, sensitivity; SPE, specificity.

Diagnostic value of circulating mSEPT9 and ctDNA concentration for HCC

We also analyzed the diagnostic performance of mSEPT9 and ctDNA concentration. The SEN, SPE, PLR, NLR, DOR, AUC, and lambda values of mSEPT9 were 0.80 (95% CI [0.71–0.87], I2 statistic:85.27%), 0.77 (95% CI [0.68–0.85], I2 statistic:80.97%), (Fig. 7), 3.57 (95% CI [2.29–5.56], I2 statistic:74.47%), 0.25 (95% CI [0.15–0.42], I2 statistic:85.67%), 14.06 (5.64–35.05, I2 statistic:85.48%), 0.86, and 2.64 (95% CI [1.73–3.55]), (Fig. 4D), respectively. The SEN, SPE, PLR, NLR, DOR, AUC, and lambda values of ctDNA concentration were 0.63 (95% CI [0.56–0.70], I2 statistic:1.38%), 0.86 (95% CI [0.74–0.93], I2 statistic:83.85%) (Fig. 8), 4.61 (95% CI [2.50–8.48], I2 statistic:57.20%), 0.42 (95% CI [0.36–0.50], I2 statistic:0.00%), 10.86 (5.60–21.07, I2 statistic:47.79%), 0.73, and 2.01 (95% CI [1.48–2.55]) (Fig. 4E), respectively.

Figure 7 Forest plots of SEN and SPE for diagnostic value of ctDNA assay for HCC in the subgroup of SEPT9 methylation.

CtDNA, circulating tumor DNA; HCC, hepatocellular carcinoma; SEN, sensitivity; SPE, specificity.

Figure 8 Forest plots of SEN and SPE for diagnostic value of ctDNA assay for HCC in ctDNA concentration detection subgroup.

Subgroup and meta regression analyses

Subgroup analysis was applied based on different covariates to explore the potential sources of heterogeneity: region (Asia vs. Africa), sample source (plasma vs. serum), control type (benign disease vs. healthy controls), sample size (≥100 vs. <100), publication year (2000–2010 vs. 2011–2021), assay methods (RT-qPCR vs. other methods in the quantitative study; MSP vs. other methods in the qualitative studies)(Table 1), and single gene methylation vs. single gene mutation in qualitative studies. It should be noted that the studies that did not distinguish patients from healthy controls were not enrolled in the control type subgroup analysis, and the number of studies in Europe (n = 2) and America (n = 3) was not sufficient enough to perform subgroup analysis. The qualitative analysis of the ctDNA subgroup based on different regions revealed that the sampling area from Asia showed better diagnostic performance (DOR:8.83, AUC:0.77) than the area in Africa (DOR:3.22, AUC:0.66). Another subgroup analysis associated with years suggested that studies from 2000–2010 (DOR:26.83, AUC:0.75) had SEN and SPE values of 0.39 and 0.98, respectively, and had lower sensitivity but greater specificity than studies from 2011–2021 (DOR:7.40, AUC:0.79), which had SEN and SPE values of 0.54 and 0.86, respectively. As for sample source, when we compared the samples collected from plasma, the sample source from serum did not show a great difference with a DOR of 5.77 vs 10.61 and an AUC of 0.76 vs. 0.77, which were quite different from the results in the quantitative subgroup. Also, studies with a sample size of ≥100 cases (DOR:7.80, AUC:0.78) were not drastically different when compared with studies with a sample size of <100 cases (DOR: 10.99, AUC: 0.78). Additionally, the single-gene mutation subgroup (DOR:5.11, AUC:0.53) had rather low SEN and higher SPE when compared with the single-gene methylation subgroup (DOR:9.05, AUC:0.79). Similarly, the quantitative analysis of the ctDNA subgroup showed a SEN of 0.65 and a SPE of 0.92, and sampling from plasma (DOR:20.54, AUC:0.87) achieved a greater diagnostic value compared to the subgroup that sampled from serum (DOR:8.00, AUC:0.79), which had a SEN of 0.71 and SPE of 0.76. Likewise, subgroup analyses related to control type showed that studies had satisfactory diagnostic value in discriminating HCC patients from healthy volunteers (DOR:59.26, AUC:0.91) compared with those using benign liver disorder patients. In terms of assay methods, research utilizing RT-PCR detective methods (DOR:22.69, AUC:0.88) revealed superior diagnostic accuracy in discriminating HCC patients from the control group with a SEN of 0.75 and SPE of 0.88 compared with research using other detective methods (DOR:10.20, AUC:0.79) with a SEN of 0.68 and SPE of 0.83, respectively.

Table 1 Subgroup analysis of the diagnostic performance of ctDNA assay for HCC.

Analysis	Group	Subgroup	SEN (95% CI)	SPE (95% CI)	DOR (95% CI)	AUC	I2 (%)	P	
Qualitative analysis	Region	Asia	0.48(0.42–0.55)	0.90(0.86–0.93)	8.83(6.20–12.48)	0.77	85.0%	0.000	
		Africa	0.37(0.16–0.63)	0.85(0.60–0.95)	3.22(1.05–9.82)	0.66	88.3%	0.000	
	Control type	HC	0.45(0.38–0.53)	0.94(0.94–0.99)	30.87(14.87–64.11)	0.79	59.6%	0.000	
		BD	0.52(0.46–0.59)	0.87(0.82–0.90)	7.06(5.14–9.69)	0.77	83.4%	0.000	
	Sample source	plasma	0.44(0.34–0.55)	0.88(0.81–0.92)	5.77(3.26–10.20)	0.76	86.0%	0.000	
		serum	0.54(0.48–0.61)	0.89(0.85–0.93)	10.61(7.54–14.94)	0.78	74.9%	0.000	
	Publication year	2000–2010	0.39(0.29–0.50)	0.98(0.92–0.99)	26.83(7.59–96.87)	0.75	48.4%	0.010	
		2011–2021	0.54(0.46–0.61)	0.86(0.82–0.90)	7.40(5.11–10.74)	0.79	89.1%	0.000	
	Sample size	≥100	0.50(0.41–0.59)	0.89(0.83–0.92)	7.80(5.42–11.22)	0.78	90.1%	0.000	
		<100	0.49(0.40–0.58)	0.92(0.86–0.96)	10.99(5.37–22.50)	0.78	75.4%	0.000	
	Assay methods	MSP	0.49(0.43–0.55)	0.90(0.85–0.93)	8.58(6.07–12.12)	0.75	72.8%	0.000	
		Other methods	0.51(0.37–0.64)	0.90(0.83–0.94)	9.56(4.84–18.88)	0.82	89.6%	0.000	
	Ctdna assay	methylation	0.52(0.45–0.58)	0.89(0.86–0.92)	9.05(6.50–12.60)	0.79	85.3%	0.000	
		mutation	0.21(0.08–0.46)	0.95(0.60–1.00)	5.11(0.41–62.71)	0.53	88.9%	0.000	
Quantitative analysis	Region	Asia	0.68(0.61–0.74)	0.85(0.78–0.91)	12.39(7.40–20.74)	0.81	86.5%	0.000	
		Africa	0.72(0.61–0.81)	0.78(0.59–0.90)	9.39(5.47–16.11)	0.80	60.9%	0.025	
	Control type	HC	0.72(0.52–0.86)	0.96(0.82–20.99)	59.26(20.24–173.49)	0.91	60.7%	0.026	
		BD	0.70(0.63–0.75)	0.8(0.72–0.87)	9.45(5.94–15.04)	0.80	79.8%	0.000	
		≥100	0.62(0.49–0.73)	0.84(0.75–0.90)	8.63(5.09–14.65)	0.81	87.6%	0.000	
		<100	0.66(0.55–0.76)	0.92(0.82–0.97)	21.93(11.11–43.29)	0.84	0.0%	0.472	
	Sample source	plasma	0.65(0.56–0.74)	0.92(0.87–0.95)	20.54(11.45–36.84)	0.87	78.6%	0.000	
		serum	0.71(0.64–0.78)	0.76(0.66–0.84)	8.00(5.15–12.43)	0.79	75.9%	0.000	
	publication	2000-2010	0.75(0.54–0.89)	0.76(0.65–0.84)	9.37(4.45–19.77)	0.81	72.3%	0.000	
		2011-2021	0.68(0.62–0.73)	0.85(0.77–0.91)	12.45(7.67–20.19)	0.80	85.5%	0.013	
	Assay methods	Rt-PCR	0.75(0.56–0.88)	0.88(0.71–0.95)	22.69(12.20–42.21)	NA	0.0%	0.832	
		Other methods	0.68(0.62–0.73)	0.83(0.75–0.89)	10.20(6.33–16.42)	0.79	86.2%	0.000	
Notes.

Abbreviations ctDNA circulating tumor DNA

HCC hepatocellular carcinoma

95% CI 95% confidence interval

DOR diagnostic odds ratio

AUC area under the curve

HC healthy controls

BD benign live diseases

MSP methylation-specific polymerase chain reaction

RT-qPCR real-time quantitative polymerase chain reaction

SEN sensitivity

SPE specificity

We also performed a multivariable meta-regression to further explore the source of heterogeneity (Table 2). The results indicated that the study region and control type may be the source of heterogeneity in the qualitative analysis subgroup. Meanwhile, none of the study characteristics shown above generated significant heterogeneity in the quantitative analysis group.

Table 2 Meta-regression of impacts of study characteristics on diagnostic performance of ctDNA for HCC.

Analysis	Covariates	Coefficient	SE	P value	
Qualitative analysis	region	1.83	0.41	0.00	
control type	−1.48	0.32	0.00	
year	−0.65	0.45	0.15	
sample size	−0.50	0.40	0.21	
assay methods	0.36	0.40	0.37	
sample source	−0.05	0.37	0.89	
ctDNA assay	−1.15	0.81	0.15	
Quantitative analysis	region	1.02	0.90	0.26	
control type	0.02	0.63	0.97	
year	0.42	1.04	0.69	
sample size	−0.65	1.00	0.52	
assay methods	−0.13	1.16	0.91	
Sample source	−0.66	0.84	0.43	
Notes.

Abbreviations 95% CI 95% confidence interval

ctDNA circulating tumor DNA

HCC hepatocellular carcinoma

SE standard error

Clinical effect

Based on Fagan’s nomogram, when we set the pretest probability to 25%, ctDNA increased the probability of a positive value to 62%, while there was a 16% probability that it ignored HCC patients with a negative test (Fig. 9A). Similarly, based on a 50% pretest probability, the probability of a correct detection increased to 83% after a 36% probability of a negative test result (Fig. 9B). When setting the pretest probability to 75%, the probability of a positive detection increased to 94%, and the probability of HCC patients being ignored increased to 63% (Fig. 9C). In the quantitative subgroup, the posttest probability increased to 59%, 81%, and 93% when we set the pretest probability to 25%, 50%, and 75% respectively, with lower post negative test results in discriminated HCC patients when compared with the qualitative analysis subgroup. Therefore, ctDNA may help AFP and ultrasounds initially screen for HCC.

Figure 9 Fagan’s nomogram for clinical utility.

Qualitative subgroup:(A) Fagan’s nomogram with 25% pretest probability. (B) Fagan’s nomogram with 50% pretest probability. (C) Fagan’s nomogram with 75% pretest probability. Quantitative subgroup: (D) Fagan’s nomogram with 25% pretest probability. (E) Fagan’s nomogram with 50% pretest probability. (F) Fagan’s nomogram with 75% pretest probability.

Publication bias

We utilized Deek’s funnel plot asymmetry test to research publication bias in the included studies (Fig. S1A & S1B). Our results showed that there was no significant publication bias in the quantitative analysis group with a coefficient of −1.90 (95% CI [−24.12–20.31]; p = 0.86), while a coefficient of 11.80 (95% CI [1.70–21.90], p = 0.023) indicated a study bias among studies that used a qualitative methodological approach. Trim and fill analysis (Chan et al., 2008) was used to correct funnel plot asymmetry from publication bias. The log OR was used as the effect estimate to execute the test. The pooled log OR was 0.685 (95% CI [0.648–0.723], p = 0.00) for the mixed model before we filled in the missing studies and the outcome altered to 0.668 (95% CI [0.632–0.705], p = 0.00) (Fig. S2). Also, we did not find publication bias in the group of ctDNA combined with AFP assay.

Discussion

HCC patients can benefit from early-stage diagnosis, but the damage and cost of tissue biopsy is not widely accepted by many patients with no or mild symptoms. Therefore, an increasing number of novel and available biomarkers for the early detection and diagnosis of HCC have been widely studied. Due to the development of next generation sequencing and other detective methods, many gene models (Cai et al., 2019) and gene panels (Li et al., 2020b) based on DNA, RNA, AFP, age, and other influencing factors have demonstrated extremely high diagnostic value. However, because of the lack of large cohort studies and the high cost of these tests, these tests cannot often be applied to clinical work. This updated meta-analysis aimed to systematically evaluate the diagnostic value of ctDNA according to the past 21 years of published results and assess the diagnostic performance of ctDNA concentration and SETP9 methylation.

The pooled SEN and SPE values based on 67 studies in the qualitative subgroup were 0.50 and 0.90, respectively, while the quantitative analysis group yielded higher SEN and SPE values of 0.69 and 0.84, respectively. The superior diagnostic performance of the quantitative analysis compared to the qualitative subgroup may be due to the low detection rate of some genes in the earlier period (Wong et al., 2000; Chang et al., 2008; Zhang et al., 2013) , low diagnostic value of single gene mutation, and advantage of selected genetic loci in non-HCC patients (Wu et al., 2017). Additionally, we concentrated on mSEPT9 methylation, which was widely used as an assay indicator in gastrointestinal cancer and frequently studied over the last two years (Oussalah et al., 2018; He et al., 2020; Kotoh et al., 2020; Li et al., 2020a; Lewin et al., 2021) , as well as circulating tumor DNA level. We found that mSEPT9 methylation discriminated HCC patients from liver cirrhosis patients and benign disease patients with a SEN of 0.80, SPE of 0.77, and AUC of 0.86. These satisfactory results suggested that mSEPT9 may have the potential to become a novel biomarker to screen HCC in clinical work. Additionally, six studies (El-Shazly et al., 2010; Chen et al., 2012; Huang et al., 2012; Chen et al., 2013; Gai et al., 2018; Linlin et al., 2018) between 2010–2020 were enrolled to evaluate the diagnostic performance of ctDNA (AUC:0.73) concentration with a SEN of 0.63 and SPE of 0.86. Our results also showed that the diagnostic value of combined ctDNA and AFP assay (AUC:0.89) distinctly increased with a SEN of 0.82 and SPE of 0.84. As a conventional biomarker in serum, AFP achieved a SEN of 0.61 and SPE of 0.86 at the threshold of 20–100 ng/mL (AUC:0.83) after enrolling 46 studies (Zhang et al., 2020a). Due to the lack of SEN and SPE, however, using AFP testing in the diagnosis of early HCC is still not ideal. In the early stage of HCC progression, the detection rate is as low as 1/3 (Wang & Wei, 2020). The quantitative and qualitative analyses of ctDNA were less sensitive but more specific compared to AFP and some gene methylation (e.g., mSEPT9 methylation) has better diagnostic value than AFP.

More importantly, we also used DOR as a single indicator to evaluate the diagnostic performance of the enrolled publications. Generally, a DOR value of >10 is considered good discriminatory performance. In this meta-analysis, the DOR values for the quantitative and qualitative ctDNA assay to distinguish HCC cases from control subjects were 11.78 and 8.72, respectively, indicating that quantitative assays showed a better performance than qualitative assays. The pooled DOR values for the quantitative and qualitative analysis of ctDNA to distinguish HCC from healthy volunteers and benign patients was 30.87 vs 7.06, and 59.26 vs 9.45, respectively. The plasma’s DOR was 20.54 in the quantitative ctDNA assay, and the serum’s was only 8.00. In addition, serum samples generally yield more ccfDNA, but the additional material is derived from leukocyte lysis during clotting, which dilutes the ctDNA content (Donaldson & Park, 2018). The DOR values of mSEPT9 and ctDNA concentration were 14.06 and 10.86, respectively, which suggested satisfactory diagnostic performance. However, it should be noted that mSEPT9 is more frequently observed in older patients (n>50) and is not a specific biomarker for HCC, suggesting that it is not sufficient for clinical use. The DOR value of the AFP and ctDNA combined assay was 23.62, while the DOR of AFP was 10.64 at the cut-off value of 20–100 ng/ml in Zhang et al. (2020a), indicating that the combined AFP and ctDNA assay exhibited dramatically powerful diagnostic performance compared to using AFP or ctDNA alone.

LRs are indicators that reflect the authenticity of SEN and SPE. Although the results of AUC and DOR suggested a high level of accuracy, our pooled PLR and NLR results were less than satisfactory. In our qualitative meta-analysis, the PLR was 4.89 and the NLR was 0.56. Our results indicated that HCC patients had approximately four to five times greater chance of a TP than the control group according to the positive test results. The NLR was 0.56, which revealed that ctDNA-negative participants may have a 56% possibility of verifying HCC. Likewise, the pooled PLR and NLR of the quantitative analysis were 4.36 and 0.37, respectively. These results were quite similar to previous studies (Zhang et al., 2020b; Zhang et al., 2021) . The poor PLR probability was not enough to support the diagnosis of HCC, and the even worse NLR suggested we should combine other biomarkers or image methods to exclude the diagnosis of HCC. Additionally, the PLR and NLR values of mSEPT9 were 3.57 and 0.25, respectively, and 4.61 and 0.42 when they related to ctDNA concentration. The addition of AFP, however, enhanced the accuracy and robustness, with a PLR of 5.13 and NLR of 0.22.

There was no published bias in the quantitative analysis and AFP-ctDNA combined group according to the asymmetric Deek’s funnel plot test. However, there were some concerns about publication bias in the qualitative analysis subgroup. Results may be biased because positive results are more likely to be published. However, the results were robust after trim and fill analysis was utilized. SEN analysis in the mSEPT9 and ctDNA subgroup was robust (Fig. S3), indicating that the results were credible. Furthermore, meta-regression analysis revealed that the covariates of study region and control type may be the sources of heterogeneity in the qualitative analysis subgroup. According to subgroup analysis, assay methods may be the source of heterogeneity in ctDNA concentration subgroup (Fig. S4). None of the study characteristics we analyzed in the quantitative subgroup presented primary heterogeneity. In fact, many factors such as vascular invasion, patient average age, tumor size, and TNM staging may cause heterogeneity, but were not taken into consideration in this study due to missing information.

During our study, we noticed genetic deviations such as mutations in TP53 (Marchio et al., 2019; Lleonart et al., 2005; Marchio et al., 2018), CTNNB1 (MacDonald, Tamai & He, 2009), and TERT (Yang et al., 2011; Akuta et al., 2021), which have been widely studied in circulating tumor DNA in HCC patients. However, a recent study identified TTN, TMEM141, UBB, and ADGRV1 also as the most frequently mutated genes in HCC patients, making them worthy of further investigation (Gao et al., 2021). In addition, genetic mutations are related to different HCC risk factors. TP53 is associated with HBV infection and aflatoxin, while CTNNB1 is mainly related to alcohol intake (Gao et al., 2021). We also found circulating cell-free DNA fragmentomics, which includes the measurement of cfDNA length and short nucleotide motifs at the ends of cfDNA molecules, provides another method of cancer diagnosis. The fragment size distribution showed a prominent peak at about 167 bp for HCC patients, HBV carriers, and healthy controls, which indicates that most of the circulating DNA molecules were derived from apoptosis (Jiang et al., 112; Jin et al., 2021). When fragment size <150 bp, fractional concentrations of tumor DNA in plasma increased in HCC patients (Jiang et al., 112; Jin et al., 2021; Meng et al., 2021) with periodic peaks and troughs in the 80- to 150-bp size range (Meng et al., 2021) observed. On the other hand, compared with non-HCC subjects, the frequencies of the 4-mer end motifs CCCA, CCAG, and CCTG significantly decreased in HCC patients with or without HBV infection, while the associations of motifs TAAA, AAAA, and TTTT with HCC is still being disputed (Jin et al., 2021; Jiang et al., 2020). These ctDNA characteristics may help guide the purification of ctDNA from cfDNA and enhance the detection rate in future studies.

The major limitations of this meta-analysis are as follows: First, in order to enroll all the studies that met the inclusion and exclusion standards as closely as possible, we took into consideration studies that included many genes with low SEN and SPE, but may have underrated the diagnostic performance of ctDNA. Moreover, the number of publications included in the ctDNA concentration subgroup and mSEPT9 subgroup were relatively small, and we need larger cohort studies to support our results. Third, most enrolled studies did not clearly point out the study type and many studies were case-control studies, which reduced the persuasiveness of the article. Also, many studies failed to provide information about some covariates such as vascular invasion, tumor staging, number of metastases, etiology, average age of participants, and tumor size. Further, the detection of samples lacks a standardized detective technology process, which may be source of heterogeneity. Finally, enrolled papers were limited to English, which may have generated some bias. Therefore, large-scale prospective studies using standardized detective technology and processes are needed in the future to support the conclusions of this meta-analysis.

Conclusion

In summary, this meta-analysis showed that quantitative and qualitative subgroups had a medium to high level of diagnostic value, and the quantitative analysis showed better diagnostic value. We also specifically analyzed ctDNA level and mSEPT9 assay, which have potential to be applied as effective novel biomarkers for HCC in clinical work. It is worth noting that the mSEPT9 assay showed a satisfactory result in a study published in 2018 (Oussalah et al., 2018). The combined assays of ctDNA and AFP yielded relatively better diagnostic performance, indicating that using ctDNA combined with conventional biomarkers may be an effective method to enhance the detection rate of HCC in the early stages. Therefore, large sample prospective studies with standardization are needed to further verify our conclusion.

Supplemental Information

Supplemental Information 1 PRISMA checklist

Click here for additional data file.

Supplemental Information 2 Basic characteristics of qualitative subgroup

Click here for additional data file.

Supplemental Information 3 Basic characteristics of quantitative subgroup

Click here for additional data file.

Supplemental Information 4 Characteristics of the included studies

Click here for additional data file.

Supplemental Information 5 Funnel plots to evaluate the publication bias for the qualitative detection subgroup and quantitative detection subgroup

(A) The qualitative detection subgroup. (B) The quantitative detection subgroup

Click here for additional data file.

Supplemental Information 6 Trim and fill analysis for qualitative studies

Click here for additional data file.

Supplemental Information 7 Sensitivity analysis: subgroup of (A) mSETP9 and (B) ctDNA concentration

Click here for additional data file.

Supplemental Information 8 Subgroup analysis of CtDNA concentration based on detection methods

Click here for additional data file.

Supplemental Information 9 The rationale for conducting the systematic review

Click here for additional data file.

We thank Mrs. Yangyu Zhang for her technical guidance and support.

Additional Information and Declarations

Competing Interests

Author Contributions

Data Availability

The authors declare there are no competing interests.

Jia Jie Li performed the experiments, analyzed the data, prepared figures and/or tables, authored or reviewed drafts of the article, and approved the final draft.

Yanqing Lv performed the experiments, analyzed the data, prepared figures and/or tables, and approved the final draft.

Huifan Ji conceived and designed the experiments, authored or reviewed drafts of the article, and approved the final draft.

The following information was supplied regarding data availability:

The raw measurements are available in the Supplemental Files.

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
