# Peer review of "Diagnostic performance of circulating tumor DNA as a minimally invasive biomarker for hepatocellular carcinoma: a systematic review and meta-analysis"

_PeerJ, doi:10.7717/peerj.14303_

## Round 0.1 · original submission · Major Revisions

Thank you for the paper. As the reviewers have suggested, please revise the paper and reply to questions.

·

Basic reporting

Dear authors,

Your manuscript updates the previous meta-analysis performed on hepatocellular carcinoma (HCC), including the last published article in 2019. After including nearly 60 studies, you had higher sensitivity and specificity values of ctDNA and AFP for classifying patients with HCC. Besides including information for the last three years, you evaluated the impact of methylated SEPTIN9 as a new candidate to be tested alone or in combination with AFP and ctDNA.

I think this submission meets the PeerJ criteria for publication. However, I would like to suggest a grammar revision and text organization to avoid confusion for the reader. For example, the "independently" term is repeated in lines 117 and 118. Then line 244 and Tables 1 and 2 show an inconsistency. Was the comparison "Asia vs. Non-Asia" or "Asia vs. Africa"?

Experimental design

no comment

Validity of the findings

no comment

Reviewer 2 ·

Basic reporting

The authors demonstrated that ctDNA can be used to improve the current diagnosis of HCC. The author provided sufficient background and raw data to support their conclusion.

There are some grammar or written English problems in the manuscript that need to be corrected before it can be accepted.

For example: in the abstract, "6 studies were included to evaluate the SETP9 methylation yielded", "6" need to be replaced by "Six".

There are also some formatting issues that need to be corrected, fo example, in Figures 3 and 4, I-square is labeled as I2. Figure 8 is missing some figure legends.

Experimental design

The methods are clearly stated with sufficient detail and information.
The research question is well defined.

Validity of the findings

The author explained the specificity and sensitivity of diagnosing HCC using ctDNA in different groups. All data has been provided, and appropriate statistical analyses were used to draw the conclusion.

Reviewer 3 ·

Basic reporting

Li et al presents a meta-analysis on circulating tumor DNA in liver cancer and proposed that ctDNA can be used as a liver cancer biomarker.

Experimental design

Overall, I find that the statistical analyses in this paper are stringent and of good quality, and the manuscript is well written. However, I do have a few major points on validity of the findings. Please see my comments in the next section.

Validity of the findings

Major point 1. It is true that ctDNA and mSEPT9 separate benign liver diseases from malignancy, but neither ctDNA nor mSEPT is specific to liver cancer. Methylated SEPT9 has been established as a marker for colorectal cancer (PMID: 31088046). And ctDNA is present in many types of cancers. The lack of specificity of ctDNA and mSEPT9 to liver cancer caner weakens authors’ claim of using them as biomarkers for liver cancer. I see this as a major weakness of this manuscript.

Major point 2. The manuscript will be more influential to a broader audience (e.g. molecular biologists, cancer biologists, computational biologists), if sequence information of ctDNA is reviewed. What is the length distribution of ctDNA seen in liver cancer patients? Is there any sequence enrichment in certain genes or genomic elements, such as LINE-1? What mutations (mentioned in line156) have the strongest correlation with liver cancer? I suggest that the authors add these analyses.

Major point 3. On line 232 mSEPT9 is brought in without much context. I suggest that the authors add a few sentences to introduce SEPT9 and its methylation in cancer.

Additional comments

Minor points
Line 302, its should be their.
Line 320, heart-stirring is a strong adjective, it should be toned down.

Reviewer 4 ·

Basic reporting

High levels of cfDNA is found in the serum or plasma of cancer patients. ctDNA (circulating tumor DNA) is a promising biomarker for a variety of cancers. Numerous studies have been conducted to evaluate the usefulness of ctDNA as a diagnostic marker in various cancer patients. In this paper, the authors did a systemic literature review to assess the diagnostic values of circulating tumor DNA, methylated SEPTIN9, and AFP in hepatocellular carcinoma (HCC). They incorporated 59 qualified studies into this meta-analysis to assess the diagnostic accuracy of circulating tumor DNA (ctDNA) in hepatocellular carcinoma (HCC). AFP is the classical HCC biomarker that is often used in diagnosing HCC. But the AFP test has exhibited unsatisfactory diagnostic performance due to its low sensitivity. When compared to ctDNA or AFP alone, mSEPT9 has been demonstrated to be more sensitive, and it has the potential to become a new HCC biomarker, especially when paired with the ctDNA assay. The combined AFP assay with ctDNA, on the other hand, significantly improved HCC diagnostic accuracy when compared to either assay alone and could be utilized as a tool for early diagnosis of HCC.

Overall, the writers communicated the literature data in a clear and simple manner. The English language and writing skills, on the other hand, must be significantly improved. Throughout the text, there were various grammatical errors. Authors must take writing seriously and devote sufficient time to evaluating their work before submitting it. Having this many errors/typos in a document demonstrates carelessness and a lack of sincerity in presenting their work.

Below are just a few of them:

Lines 4-9: It seems all three author affiliations are basically the same. In that case, you can use just one lower-case superscript number for all the authors.
Line 23: It is not mSETP9, but rather mSEPT9. Also, maintain consistency in writing the abbreviations throughout the text.
Line 86: not a full stop. Use a semicolon.
Line 150, 194, 211, 247, 258, 293, 294, 297,……..: Use a space between sentences and in front of the opening parenthesis. Use commas where necessary.
Line 177: Use a comma instead of a full stop.
Line 210: "symmetrical"
Line 226: No need to write "respectively" here.
Line 228: Write "respectively" here.
Line 257: Concern sample source……… Rewrite this sentence.
Lines 311 and 313: Maintain consistency in writing abbreviations for sensitivity and specificity. (SEN, SPE) or (SE, SP)?
Line 314: "Quantitative" is used twice.
Line 357: excluded?
Line 369: Use a full stop between sentences.
Line 377: msept9 (mSEPT9).

Figure captions were not complete for Figures 1, 2, 8, and 9.
Figure 2: In the text (line 153), the authors mentioned that they included studies from 59 papers. But in Figure 2, they listed 60 papers.
Write captions for supplemental figures.

Experimental design

No comment

Validity of the findings

No comment

---

## Round 0.2 · Minor Revisions

The paper has been greatly improved. However, a minor revision is still needed.

Reviewer 3 ·

Basic reporting

The manuscript by Li et al presents a meta-analysis on cell free circulating tumor DNA, in particular, mSEPT9, and their potential use as liver cancer diagnostic markers. The authors have addressed or provide adequate response to my comments. While I still believe that the sequence information of mSEPT9 is crucial, I understand that it might be beyond the scope of this paper. The writing of this manuscript has been improved but there are still several omissions that would benefit from being addressed.

Experimental design

I find that the statistical analyses in this paper are stringent and of good quality.

Validity of the findings

Line 126-127 (corresponding to Figure 2). In the categorization of risk of bias and applicability, how are “low”, ”unclear”, and ”high” defined? There should be an explanation on the categorization criteria.

Figure 4. Labeling of subpanels (A, B, C, etc) should be in the top left corner of each panel.

---

## Round 0.3 · accepted · Accept

The paper can be accepted.